Temporal dynamics of the developing lung transcriptome in three common inbred strains of laboratory mice reveals multiple stages of postnatal alveolar development

Beauchemin Kyle J. 1 2
Wells Julie M. 1
Kho Alvin T. 3
Philip Vivek M. 1
Kamir Daniela 1
Kohane Isaac S. 4
Graber Joel H. 1
Bult Carol J. carol.bult@jax.org 1
1 The Jackson Laboratory , Bar Harbor , ME , United States
2 Graduate School of Biomedical Sciences and Engineering, The University of Maine , Orono , ME , United States
3 Computational Health Informatics Program, Boston Children’s Hospital , Boston , MA , United States
4 Department of Biomedical Informatics, Harvard Medical School , Boston , MA , United States
Heath Joan
Electronic publication date: 2016 Aug 9
Publication date: 2016
Volume: 4
Electronic Location ID: e2318
Received 2016 May 13; Accepted 2016 Jul 12
Copyright: ©2016 Beauchemin et al.
Copyright year: 2016
Copyright holder: Beauchemin et al.
License: This is an open access article distributed under the terms of the Creative Commons Attribution License, which permits unrestricted use, distribution, reproduction and adaptation in any medium and for any purpose provided that it is properly attributed. For attribution, the original author(s), title, publication source (PeerJ) and either DOI or URL of the article must be cited.
License URL: https://creativecommons.org/licenses/by/4.0/

Keywords: Lung development, Genomics, Gene expression, Laboratory mice, Time series

Funding: The Jackson Laboratory (JAX) Faculty Development Fund The Maine Cancer Foundation NIH HD068250 P30CA034196 HL91124 The research reported in this publication was funded in part by The Jackson Laboratory (JAX) Faculty Development Fund, the Maine Cancer Foundation, NIH HD068250, P30CA034196, and HL91124. The funders had no role in study design, data collection and analysis, decision to publish, or preparation of the manuscript.

==============================
To characterize temporal patterns of transcriptional activity during normal lung development, we generated genome wide gene expression data for 26 pre- and post-natal time points in three common inbred strains of laboratory mice (C57BL/6J, A/J, and C3H/HeJ). Using Principal Component Analysis and least squares regression modeling, we identified both strain-independent and strain-dependent patterns of gene expression. The 4,683 genes contributing to the strain-independent expression patterns were used to define a murine Developing Lung Characteristic Subtranscriptome (mDLCS). Regression modeling of the Principal Components supported the four canonical stages of mammalian embryonic lung development (embryonic, pseudoglandular, canalicular, saccular) defined previously by morphology and histology. For postnatal alveolar development, the regression model was consistent with four stages of alveolarization characterized by episodic transcriptional activity of genes related to pulmonary vascularization. Genes expressed in a strain-dependent manner were enriched for annotations related to neurogenesis, extracellular matrix organization, and Wnt signaling. Finally, a comparison of mouse and human transcriptomics from pre-natal stages of lung development revealed conservation of pathways associated with cell cycle, axon guidance, immune function, and metabolism as well as organism-specific expression of genes associated with extracellular matrix organization and protein modification. The mouse lung development transcriptome data generated for this study serves as a unique reference set to identify genes and pathways essential for normal mammalian lung development and for investigations into the developmental origins of respiratory disease and cancer. The gene expression data are available from the Gene Expression Omnibus (GEO) archive (GSE74243). Temporal expression patterns of mouse genes can be investigated using a study specific web resource (http://lungdevelopment.jax.org).

Introduction

Proper development of the mammalian respiratory system requires the spatiotemporal coordination of molecular interactions among more than 40 different cell types (Breeze & Wheeldon, 1977) to form a complex, highly branched structure and associated vasculature for facilitating gas-exchange. Although there are differences between mouse and human lung anatomy and cell type distribution (Braun et al., 2012; Irvin & Bates, 2003; Keli, Mogha & Egan, 2010; Wright, Cosio & Churg, 2008), the basic morphological hallmarks of the developing lung are comparable between mouse and human (Rackley & Stripp, 2012). These conserved features make the laboratory mouse an invaluable model system for identifying and characterizing genes, pathways, and networks that are fundamental to normal lung development and disease in humans (Dutt & Wong, 2006; Moore et al., 2013; Rawlins & Perl, 2012; Wright, Cosio & Churg, 2008).

The process of mammalian lung development is traditionally described as five stages defined by histological features, cell type composition, and morphology: embryonic, pseudoglandular, canalicular, saccular, and alveolar (Have-Opbroek, 1991; Warburton et al., 2000) (Fig. 1). In mice, lung development initiates during the embryonic stage (EMB) with the emergence of lung buds from the ventral foregut endoderm and subsequent septation from the esophagus. Several growth factors and transcription factors including NK2 homeobox 1 (Nkx2-1), GATA binding protein 6 (Gata6), GLI-Kruppel family members 2 and 3 (Gli2/3), Sonic hedgehog (Shh), and Fibroblast growth factor 10 (Fgf10) are key molecular drivers of this initiation phase (Lu et al., 2005; Millien et al., 2008; Warburton et al., 2005; Weaver, Dunn & Hogan, 2000). The pseudoglandular stage (PSG) is characterized by stereo-specific branching morphogenesis of the lung bronchi and the formation of pre-alveolate saccules (Herriges et al., 2012; Metzger, Stahlman & Shannon, 2008). Branching morphogenesis is an iterative process of bud growth, elongation, and subdivision of terminal units, which ultimately generates the bronchial tree, complete with secretory gland, blood vessels, and inter alveolar septa (Lu et al., 2004). Genes that underlie branching morphogenesis include members of several well-known developmental signaling pathways (e.g., wingless-type MMTV integration site family (WNT), FGF, and transforming growth factor beta (TGFb) as well as homeobox-containing, zinc finger, forkhead box-containing, SRY-containing, and ETS domain containing genes (Herriges et al., 2012; Lu et al., 2004; Minoo et al., 1999). The canalicular stage (CAN) follows the formation of the terminal bronchioles of the five lung lobes. This stage is defined by expansion of the canaliculi, which form the pulmonary parenchyma, and concurrent expansion of the surrounding capillary network. This vascularization of the lungs is largely driven by interdependent vascular endothelial growth factor (VEGF) signaling between the vascular endothelium and pulmonary epithelium (Warburton et al., 2010). Dramatically increased vascularization and establishment of saccules (primitive alveoli) within the canaliculi characterizes the start of the saccular stage (SAC) during which immature terminal sacs form as primitive respiratory sites poised for gas exchange following birth. Morphologically, expansion of the capillary bed continues during this stage, along with pulmonary innervation and establishment of the lymphatic network (Schraufnagel, 2010). The development of secondary septa to transform terminal sacs into mature alveolar ducts and alveoli finally occurs post-birth during the alveolar stage (ALV). There is continuing debate over the definition of the alveolar stage. Some groups have described postnatal maturation as an extension of the saccular stage of development (Warburton et al., 2010) while others have proposed two distinct phases of postnatal alveolarization (Mund, Stampanoni & Schittny, 2008; Schittny, Mund & Stampanoni, 2008).

Figure 1 Comparison of mouse strains, time points, and platforms for genome wide expression profiling of murine lung development.

Shaded cells indicate time points included in published transcriptional profiles of mouse lung development. Images reflect the major anatomical changes at each embryonic stage of development.

Several previous studies have reported on genome-wide gene expression profiling to characterize transcriptional programs active during lung development in both mouse (Bonner, Lemon & You, 2003; Kho et al., 2009; Lu et al., 2004; Mariani, Reed & Shapiro, 2002; Xu et al., 2012) and human (Kho et al., 2010) (Fig. 1). In mouse, these studies have identified genes and networks involved in branching morphogenesis (Lu et al., 2004), cell cycle/apoptosis (Bonner, Lemon & You, 2003), alveolarization/maturation (Kho et al., 2009; Xu et al., 2012) and extracellular matrix (ECM) development (Mariani, Reed & Shapiro, 2002). Treutlein et al. (2014) used single cell RNA-Seq to characterize the lung epithelium lineage hierarchy in mouse, expanding our knowledge of lung-specific epithelial cell types and the molecular markers that can be used to identify those cell types. The LungMAP project (www.lungmap.net) used “next generation” sequencing technologies to profile gene expression in the laboratory mouse from whole lung tissue and from populations of sorted cells between late canalicular and saccular stages through alveolarization. In humans, the most comprehensive transcriptomic analysis of lung development identified a novel molecular substage of the pseudoglandular stage that is not readily apparent by histology (Kho et al., 2010). Even with the discoveries of the genomic contributions to the developing lung in mouse and human enabled by these previous transcriptional surveys, gaps remain in our understanding of the temporal dynamics of gene expression during mammalian lung development, how genetic variation influences expression patterns in different strains of laboratory mice, and the degree to which the core set of lung development genes in human compare to mouse.

We report here a study of genome-wide gene expression during normal lung development and maturation in three inbred laboratory strains of mice (A/J, C57BL/6J, and C3H/HeJ). Relative to previous transcriptomics studies in the laboratory mouse, our study includes greater temporal resolution of time points during pre- and post-natal stages of murine lung development (Fig. 1) and serves as the most comprehensive genomic characterization of murine lung development to date. The three mouse strains included in our study were selected based on known differences in their normal lung physiology and respiratory disease susceptibility, including lung function (Reinhard et al., 2002; Schulz et al., 2002; Soutiere, Tankersley & Mitzner, 2004), response to radiation-induced lung damage (Jackson, Vujaskovic & Down, 2011), susceptibility to lung cancer (Gariboldi et al., 2003; Gordon & Bosland, 2009; Hoag, 1963) susceptibility to pulmonary fibrosis (Haston et al., 1996; Lemay & Haston, 2005; Walkin et al., 2013), airway responsiveness (De Sanctis & Drazen, 1997; Whitehead et al., 2003), and airway remodeling (Shinagawa & Kojima, 2003). The patterns of differential gene expression observed among these strains serves as the basis to identify developmental genomic factors that may contribute to adult respiratory disease susceptibility. The developing lung transcriptomics data described here are a unique resource for (1) identifying key transcriptional programs required for normal murine lung development, (2) understanding the genomic basis for strain specific differences in lung biology, (3) comparing genomics of lung development in mouse and human, and (4) providing insights into the relationship of normal lung development to respiratory disease processes (Desai & Cardoso, 2002; Gariboldi et al., 2003; Shi, Bellusci & Warburton, 2007; Yates & Dean, 2011).

Materials and Methods

Tissue and RNA

Embryos and mice for this study were collected from timed pregnant mice for C57BL/6J (B6; stock # 000664), A/J (AJ; stock # 000646), and C3H/HeJ (C3H; stock # 000659) inbred lines from The Jackson Laboratory. Whole lungs were surgically dissected at embryonic (E) days E11.5, E12.5, E13.5, E14.5, E15.5, E16.5, E17.5, E18.5, E19.5 (A/J only) and postnatal (P) days P0, P2, P3, P4, P5, P7, P9, P11, P12, P13, P14, P18, P21, P24, P30, and P56 (Fig. 1). Dissected tissues were placed immediately in RNALater (Ambion) and stored at -80°C. Due to the difficulty of manually dissecting lung buds at E9.5, whole embryos were harvested for RNA extraction at this time point. For time points E11.5 and E12.5, tissues were pooled from multiple embryos (four and three, respectively) to obtain sufficient RNA for array analysis. For all subsequent time points, tissues from at least three animals were collected and analyzed separately. As the A/J strain has an extra day of gestation relative to the other inbred strains used in this study, embryos were collected at E19.5 only for this strain. Only male mice were used in this study. The sex of every embryo and pup less than 21 days of age was determined by PCR of DNA extracted from tail fragments using primers designed to amplify the male-specific Sry gene (MGI:98660) (Data S1). All animal work was performed in accordance with Jackson Laboratory Animal Care and Use Committee (ACUC) protocol 101011 (CJB).

Total RNA was extracted using mirVana RNA isolation kits (Ambion) following manufacturer’s directions. An additional DNase I (Promega) digestion step was added before organic extraction. The quality of each RNA sample was assessed using RNA 6000 Nano LabChip assays and a 2100 Bioanalyzer (Agilent Technologies, Palo Alto, CA, USA). RNA purity (OD260/280 and OD 260/230 ratios) was assessed using a NanoDrop spectrophotometer (Thermo Fisher Scientific); only samples with ratios above 2.0 were included in this study.

Data processing

Gene expression was assayed using the Affymetrix Mouse Gene 1.0 ST array platform (Affymetrix). The .CEL files from the arrays were deposited in the Gene Expression Omnibus (GEO) archive at NCBI (Barrett et al., 2013) (Accession GSE74243). The array data were processed using the DeSNPer pipeline developed at the Center for Genome Dynamics (CGD) at The Jackson Laboratory (https://github.com/jaxcs/DeSNP). The DeSNPer algorithm removed probes with single nucleotide polymorphisms (SNPs) (Yalcin et al., 2012) among the three strains from further analysis to ensure that gene expression differences detected among the strains were not due to hybridization biases resulting from strain-specific sequence variation. Following probe removal, the data were log2-transformed, then probe-level hybridization intensities were quantile normalized and median polished to generate summarized gene-level expression values (Irizarry et al., 2003). A Z-score transformation was then applied, standardizing each gene to an average of zero and standard deviation of one across all samples. Finally, to focus data interpretation on the dominant patterns of gene expression changes across the developmental time series, the dataset was filtered to retain the top 30% genes with the largest pre-Z-transform standard deviation (referred to as the “filtered dataset”; Data S2). The temporal expression patterns for all of the genes measured on the arrays can be visualized using the web interface implemented for this project (http://lungdevelopment.jax.org/).

Principal Component Analysis (PCA)

The filtered dataset described above (Data S2) was analyzed using Principal Component Analysis (PCA), a data transformation method to reduce the dimensionality of a dataset to those orthogonal components that account for the maximum variance in the data with the fewest number of observations. PCA was performed using JMP® v11.1.1 (SAS Institute) using the principal component toolset within the multivariate library. All default settings were used and no weighted bias was applied. PCA was first used to identify any systematic biases between samples, specifically among batches of arrays that were processed together (i.e., batch effects). This process identified a subset of 42 arrays that were strongly correlated with each other independent of strain or time point. The batches containing these arrays corresponded with changes in reagent chemistry used for array hybridization and were removed from further analysis. PCA was then performed on the final set of 216 arrays to identify global patterns of expression variation between samples. This analysis generated two data matrices: gene loading values (Data S3), which represent the correlation between each gene and the expression patterns captured by each principal component (PC), and sample scores (Data S4), which reflect the relationship between each sample and each principal component. The gene loading values are either positive or negative, with positive (negative) values reflecting correlation (anti-correlation) between the gene expression and the sample score plot (Fig. S1). Characteristic gene sets for each component were created using the most highly correlated genes (positively and negatively) from each PC.

Regression analysis

Two standard least squares regression analyses were performed in JMP® (v11.1.1) to model the strain and developmental time point effects associated with the gene expression patterns identified by PCA. In the first analysis, the PCA score was modeled with independent additive effects from developmental time point and mouse strain: Pij = ai + bj + eij, where Pij is PCA score, ai is the effect of time point i, bj is the effect of strain j, and eij is the residual for strain j at time i. There were not sufficient data to model a strain-by-time point interaction term. Multiple hypothesis correction was implemented by controlling for FDR < 0.1 (Benjamini & Hochberg, 1995). The computed time point effects were plotted by component and used to identify contiguous bins of time points with similar PCA scores. These bins, in turn, were used to define gene expression-based stages of lung development. In cases of overlapping blocks between different block endpoints, a visual inspection was used to manually select the best grouping (e.g., postnatal time point 3 was grouped with P0 and P2, but could alternatively have been included with P4 and P5).

In the second analysis, PCA score was modeled as function of strain, stage, and the interaction of strain and stage using the stages identified in the first regression analysis: Pij = di + bj + gij + eij, where Pij is PCA score, di is the effect of developmental stage i, bj is the effect of strain j, eij is the interaction term between stage i and strain j, and eij is the residual for strain j at time i. Effect significance was corrected for by multiple hypothesis testing at FDR < 0.1, and pairwise comparisons between effects (stages, strains, and stage*strain) were carried out with the Tukey-Kramer Honestly Significant Difference (HSD) test (Kramer, 1956). The second regression analysis was performed to estimate effects for each strain (presumed to be constant across all time points), effects for each stage (presumed to be constant across all strains), and effects for each strain at each stage, allowing for the identification of stage-specific differences among strains for each PCA component.

Genes from the filtered dataset (Data S2) were binned based upon their patterns of expression as a function of stage and/or strain to identify strain-dependent differential expression. Standard linear pairwise correlation analysis (Pearson correlation) was also used to quantify relationships between strains from each PCA score (Table 1).

Table 1 Summary of analysis results for lung development transcriptional profiling in three inbred mouse strainse.

The cumulative variation in gene expression accounted for by the first 10 principal components is 80.9%. Pearson Correlation results are from the pairwise correlation of PCA scores as a function of strain. Regression results show the significance levels of the effect of three variables (developmental stage, strain, and stage by strain interaction) for each Principal Component. Tukey HSD connecting letters diagram indicates relatedness of strains for each component.

Component	Var. (%)	Pearson Correlation (R2)	Regression (Pval)	Tukey HSD	
PC1	55.1 (55.1)	C57BL/6J-A/J	.990***	Stage	<.0001†	AJ	A	
A/J-C3H/HeJ	.993***	Strain	0.9131	B6	A	
C57BL/6J-C3H/HeJ	.990***	Stage*strain	0.6566	C3H	A	
PC2	6.1 (61.2)	C57BL/6J-A/J	.744***	Stage	<.0001†	AJ	A	
A/J-C3H/HeJ	.839***	Strain	0.5970	B6	A	
C57BL/6J-C3H/HeJ	.849***	Stage*strain	0.0038	C3H	A	
PC3	4.98 (66.2)	C57BL/6J-A/J	.865***	Stage	<.0001†	AJ	A	
A/J-C3H/HeJ	.739***	Strain	0.2229	B6	A	
C57BL/6J-C3H/HeJ	.646***	Stage*strain	0.2090	C3H	A	
PC4	3.53 (69.7)	C57BL/6J-A/J	.117	Stage	<.0001†	AJ	B	
A/J-C3H/HeJ	.114	Strain	0.0072	B6	B	
C57BL/6J-C3H/HeJ	.192	Stage*strain	0.0029	C3H	A	
PC5	2.83 (72.5)	C57BL/6J-A/J	.379*	Stage	<.0001†	AJ	B	
A/J-C3H/HeJ	.445**	Strain	<.0001†	B6	C	
C57BL/6J-C3H/HeJ	.262	Stage*strain	<.0001†	C3H	A	
PC6	2.27 (74.8)	C57BL/6J-A/J	.130	Stage	<.0001†	AJ	B	
A/J-C3H/HeJ	.231	Strain	<.0001†	B6	A	
C57BL/6J-C3H/HeJ	.214	Stage*strain	0.0006†	C3H	B	
PC7	2.02 (76.8)	C57BL/6J-A/J	.522***	Stage	<.0001†	AJ	A	
A/J-C3H/HeJ	.561***	Strain	<.0001†	B6	B	
C57BL/6J-C3H/HeJ	.734***	Stage*strain	0.0385	C3H	A	
PC8	1.55 (78.4)	C57BL/6J-A/J	.286	Stage	<.0001†	AJ	B	
A/J-C3H/HeJ	.469***	Strain	<.0001†	B6	A	
C57BL/6J-C3H/HeJ	.131	Stage*strain	0.0437	C3H	A	
PC9	1.43 (79.8)	C57BL/6J-A/J	.344*	Stage	<.0001†	AJ	B	
A/J-C3H/HeJ	.479***	Strain	<.0001†	B6	C	
C57BL/6J-C3H/HeJ	.194	Stage*strain	<.0001†	C3H	A	
PC10	1.08 (80.9)	C57BL/6J-A/J	.320	Stage	<.0001†	AJ	C	
A/J-C3H/HeJ	.386**	Strain	<.0001†	B6	B	
C57BL/6J-C3H/HeJ	.181	Stage*strain	<.0001†	C3H	A	
Notes.

* p-value < 0.01.

** p-value < 0.001.

*** p-value < 0.0001.

† Significant effect after multiple testing correction (FDR < 0.1).

The dataset used as input for the PCA and regression analyses is provided in Data S2.

Differential gene expression analysis: Significance of Microarrays (SAM)

Differential gene expression between stages of development was determined using Significance Analysis of Microarrays (Tusher, Tibshirani & Chu, 2001) with a median FDR = 0 using the TM4 package (Saeed et al., 2003). Binning of individual time points was based on the developmental stages defined by the regression analysis described above.

Quantitative Real-Time PCR (qPCR) validation of strain-specific gene expression

We used quantitative Real-Time PCR (qPCR) to validate strain-specific expression of selected genes. cDNA was generated from nine samples of total RNA originally used for microarray analysis, using RT2 First Strand Kit (Qiagen) following manufacturer’s directions. cDNA was generated from three biological replicates for each strain at postnatal day 13 (P13). This time point was chosen, in part, because the strain-specific expression differences of the genes selected for validation (see below) were evident at this time and there was sufficient quantities of the same high quality RNA used for the microarray analysis to use for qPCR. Gene expression was assayed using RT2 SYBR Green ROX qPCR MasterMix Kit (Qiagen) following manufacturer’s directions. Three genes with low variation in expression by time point or strain (Actb, Rpl10, Rpl13a) were used to normalize expression values of the target genes. A geometric mean of the control genes was determined for each biological triplicate to generate a normalization value. Following normalization, relative expression values were calculated for each target gene (Fggy, Saa3, Wnt11, Wif1) within all biological groups. The genes for validation were selected because of their relevance to fibrosis-related biological processes that are known to differ among inbred strains of mice (Walkin et al., 2013). The significance of differences between strains in relative expression arbitrary units (A.U.) was assessed by Tukey Multiple testing corrected ANOVA using Prism v7.0a (GraphPad Software). Primer sequences used for qPCR were as follows: Actb, 5′-GTGACGTTGACATCCGTAAAGA-3′, 5′-GCCGGACTCATCGTACTCC-3′; Rpl10, 5′-CGTGGTGTCCCTGATGCTAAG-3′, 5′-GTTGGCACAAATACGGGCAG-3′; Rpl13a, 5′-AGCCTACCAGAAAGTTTGCTTAC-3′, 5′-GCTTCTTCTTCCGATAGTGCATC-3′; Fggy, 5′-CGGCTCTAGTGGACCAGAGA-3′, 5′-ATGTGCATCAATCCCTTGAACA-3′; Saa3, 5′-AGAGAGGCTGTTCAGAAGTTCA-3′, 5′-AGCAGGTCGGAAGTGGTTG-3′; Wnt11, 5′- ATGCGTCTACACAACAGTGAAG-3′, 5′-GTAGCGGGTCTTGAGGTCAG-3′; Wif1, 5′-CCTGCCGAAATGGAGGTAAA-3′, 5′-GCTGGCTCCATACCTCTTATTG-3′.

Annotation term enrichment analysis

Annotation enrichment analysis was performed to guide the interpretation of gene sets associated with specific developmental stages and for those contributing to individual principal components (PCs). Probe set identifiers from the Affymetrix arrays were first mapped to mouse gene identifiers using Mouse Genome Informatics (MGI) database Batch Query tool (Bult et al., 2010). Sets of mouse gene symbols or gene accession identifiers were submitted for Gene Ontology (2015) and Mammalian Phenotype Ontology (Smith & Eppig, 2012) term enrichment analysis using VisuaL Annotation Display (VLAD) v1.5.1 (Richardson & Bult, 2015). Mouse genes that were converted to human homologs were also analyzed using the Reactome Pathway Browser v52 (Croft et al., 2014) to identify enrichment of homologous pathway annotations.

Mouse and human gene homology

Gene homology between mouse and human was used to compare transcriptional profile results from this study to those from a previous study of human lung development (Kho et al., 2010). A gene homology resource containing 10,266 genes was generated manually using homology assertions from Panther (Mi, Muruganujan & Thomas, 2013) and Homologene (Sayers et al., 2012). If one gene mapped to multiple homologs, each homolog was represented in the final set. For example, the mouse alcohol dehydrogenase 1 (Adh1) gene is homologous to ADH1A, ADH1B, and ADH1C in human; the carboxylesterase (CES) gene in human is homologous to Ces1d, Ces1e, and Ces1g in mouse. Many-to-one homolog relationships were identified in 18 instances. Overlap between lists of homologous genes was assessed with the GeneVenn tool (Pirooznia, Nagarajan & Deng, 2007) using the NCBI GeneID for the human gene as the primary gene accession identifier.

Results

Principal component analysis (PCA) was used to identify global patterns of variation in gene expression across the lung development time series. Regression analyses were performed to group time points into developmental stages and also to associate the global patterns identified by PCA with specific effects of the experimental design, including strain and stage of development.

Principal component and regression analysis reveals nine stages of murine lung development

The first three principal components (PCs) accounted for 66.2% of the variation among samples; the first ten PCs accounted for 80.9% of the sample variation (Table 1). The patterns of variation in gene expression represented by PCs 1–3 represent common patterns of expression that do not very significantly between the three mouse strains (Fig. 2). In contrast, PCs 4–10 demonstrated strain-specific patterns of expression variation (Fig. S2). Regression modeling supported the PCA results (Table 1); no significance was detected among the strain or strain by stage effects for PCs 1–3 whereas PC 4–10 all were found to have differences between one or more of the strains for some of the developmental stages (Fig. 3). To identify possible temporal shifts in gene expression patterns between strains, correlations across all strain by PC combinations were performed. No significant correlations from this analysis were observed.

Figure 2 Global patterns of sample variation across lung development.

Plots of PCA scores (y-axis) for strain-independent principal components 1–3 along developmental time points and stages (x-axis). Time points: embryonic (E); postnatal (P). Stages: whole embryo (WE); embryonic (EMB); pseudoglandular (PSG); canalicular (CAN); saccular (SAC); alveolar (ALV1-4); mature lung (MAT). (A) PCA scores for principal components 1–3 (averaged across all three strains) across all developmental time points. (B) PCA scores for principal components 1–3 plotted for each mouse strain.

Figure 3 Regression modeling of gene expression as a function of strain and developmental stage.

Results of the linear regression analysis performed on PCA scores from strain-dependent principal components (PC 4–10). (A) Plots of least square means (y-axis) showing stage effects. (B) Plots of least square means (y-axis) illustrating strain effects. (C) Annotation enrichment results for characteristic gene sets with positive or negative loadings on PCs 4–10.

Regression analyses of the PCA results support the grouping of sampled time points into nine stages of lung development (Fig. 4). The four prenatal stages, embryonic (EMB, E9.5–E12.5), pseudoglandular (PSG, E13.5–E15.5), canalicular (CAN, E16.5–E17.5), and saccular (SAC, E18.5–E19.5) are concordant with those defined previously by histology and morphology. We identified four molecularly distinct stages of alveolar development between P0–P18 (ALV1-4) that are defined by the expression patterns and functional properties of differentially expressed genes. Finally, the time points following alveolarization were grouped under the common heading of mature lung (MAT, P21–P56).

Figure 4 Regression analysis supports nine molecular stages of lung development.

Each set of colored bars represents the least square means of the stage effects for principal components 1–3. A summary of annotation enrichment categories are shown for the positive and negative characteristic gene sets from each component (complete results provided in Data S6).

Strain-independent principal components 1–3 define a murine developing lung characteristic subtranscriptome (mDLCS)

The first PC (55.1% of the sample variation) was significantly correlated (P < 0.0001) with developmental time point, capturing the patterns of gene expression across the entire developmental timeline. Over 50% of the genes in our filtered dataset (Data S2) had relatively high (positive) or low (negative) loading values on PC1. GO term enrichment analysis of genes contributing to the prenatal signal (PC1pos) revealed enrichment of genes associated with nucleic acid metabolic process (GO:0090304) and RNA processing (GO:0006396). Genes previously associated with lung cell differentiation were among the top 10% of contributors to PC1 (Fig. S6); a 3.2-fold enrichment (Fisher exact test; P < 1.7 × 10−3). Annotation enrichment analysis of genes contributing to the postnatal signal (PC1neg) identified enrichment of immune system processes (GO:0002376), cellular communication (GO:0010646), and localization (GO:0051179). Specifically, we observed postnatal induction of genes associated with RAS protein superfamily (RAS), Ras-related protein 1 (Rap1), phosphatidylinositol 3-kinase/protein kinase B (PI3K/Akt), and mitogen-activated protein kinase (MAPK) signaling as well as chemokine signaling and phagocytosis/endocytosis, including seven toll-like receptors, four lymphocyte antigens, six chemokine receptor/ligands, and eighteen interleukins (Table S1). Many surfactants (4 of 4), mucins (2 of 4), and extracellular matrix genes (6 of 10 laminins; 7 of 21 collagens; 10 of 18 integrins) also contribute significantly to PC1neg.

The pattern of sample variation captured by PC2 revealed major changes in gene expression between the EMB/PSG and ALV3/ALV4 stages, depicted as shifts in PC2 sample coordinates between stages (Fig. 2). The shift between EMB/PSG stages represents a molecular transition from early embryonic organogenesis to prenatal epithelial/vascular development; genes that were significantly down regulated between these stages are primarily associated with embryonic morphogenesis whereas up regulated genes are associated with cell migration and angiogenesis. The shift at ALV3/ALV4 represents a transition from active cellular growth to homeostasis; genes that were down regulated between these stages are associated with cell cycle whereas genes that were up regulated are associated with metabolism (lipid, oxidation–reduction) and immune response.

Shifts in gene expression captured by PC3 occur between the PSG/CAN, ALV1/ALV2, and ALV4/MAT stages (Fig. 2). Down regulated genes between PSG/CAN are primarily associated with cell cycle and up regulated genes are associated with cellular differentiation and localization, suggesting that processes related to differentiation and morphogenesis dominate late prenatal development. The shift between ALV1/ALV2 also represents a transition from metabolism/homeostasis to induction of cell cycle; down regulated genes between these stages are associated with lipid metabolism, chemical homeostasis, and localization while up regulated genes are highly enriched for cell cycle. All differentially expressed genes between molecular stages can be found in Data S5.

The murine developing lung characteristic subtranscriptome

Because the sample scores for PC1-3 were significantly correlated among the three inbred strains of mouse (P < 0.0001) (Table 1), we used the genes contributing to these components to define a murine Developing Lung Characteristic Subtranscriptome (mDLCS). The mDLCS represents a strain-independent core set of genes associated with normal murine lung development. We defined the mDLCS using an approach described previously for the human lung transcriptome (Kho et al., 2010) with some modifications. Due to the bimodal distribution of gene loadings onto PC1 (Fig. 2) we included 3468 genes with the most extreme positive (25%) or negative (25%) gene loadings on this component. For PC2 and PC3, where gene loadings were more normally distributed, we included 1,396 genes with the highest positive (10%) or lowest negative (10%) loadings. The resulting set of 4,683 unique genes constitutes the mDLCS (Data S6).

GO term enrichment analysis was used to identify the biological processes represented by the genes contributing to the mDLCS. The results demonstrated significant enrichment for genes associated with general developmental processes (1,371 genes) including, tube, cardiovascular, mesenchyme and epithelium development. Among this developmentally-enriched set were 84 genes specifically annotated as being involved in lung development including, early lung formation (Nkx2-1, Rdh10) (Kimura et al., 1996; Sandell et al., 2007), branching morphogenesis (Hoxa5, Fgfr1, Fgfr2) (Boucherat et al., 2013; Hokuto, Perl & Whitsett, 2003), pulmonary vascular development/epithelial-mesenchymal crosstalk (Wnt5a, Shh, Fgf10, Fgf18) (Bellusci et al., 1997a; Bellusci et al., 1997b; Cornett et al., 2013; Usui et al., 2004), alveolarization/distal maturation (Hif3a, Errfi1, Igf1) (Huang et al., 2013; Jin et al., 2009; Pais et al., 2013), and pulmonary inflammatory response (Igfbp4, Nedd4l, Apoe, Fgfr4) (Cantin et al., 2015; Kimura et al., 2011; Massaro & Massaro, 2008; Rezvani et al., 2013). A full report of the enrichment analysis results for the mDLCS is provided in (Data S6).

To compare previously reported lung development genes in mouse with those captured by the mDLCS we compared the genes included in the mDLCS to lists of genes previously associated with lung development in the mouse (Data S8). We compiled a list of 657 genes from four previous transcriptomics studies (Bonner, Lemon & You, 2003; Lu et al., 2004; Mariani, Reed & Shapiro, 2002; Xu et al., 2012). From this gene set, 161 genes were removed because they were not assayed by the Affymetrix Mouse 1.0ST array or did not pass variance filtering in the current study. Of the 496 unique genes that remained, the mDLCS included 406 genes (82%). These genes included regulators of cellular adhesion (Cav1, Cdh1) and the neonatal immune response (Stat1, Egfr) (Xu et al., 2012), cell cycle regulation (Igf2, Mest, Peg3) (Finkielstain et al., 2009), and cellular metabolic processes associated with terminal differentiation (Elf3, Klf4) (Bonner, Lemon & You, 2003). Among the 90 genes that were not included in the mDLCS, 59 were captured by one or more principal components (PC4–10) with strain-specific patterns in gene expression associated with lung development.

We also compared the mDLCS gene list to all mouse genes currently annotated to the Gene Ontology term “lung development” (GO:0030324) and to genes annotated with the Mammalian Phenotype ontology term “respiratory system phenotype” (MP:0005388). We downloaded these annotations from the MGI database (v6.01; 12/22/15) based on MGI’s unified gene catalog (genome assembly GRCm38) (Zhu et al., 2015). The resulting list included 209 genes annotated to the GO term of lung development (GO:0030324) and its child terms and 1,076 genes annotated to the MP term “respiratory system phenotype” (MP:0005388) and its child terms; 134 genes shared both “lung development” and “respiratory system phenotype” annotations. The gene list from previous transcriptomic studies in mouse and the gene list created from GO and MP terms overlapped by 32 genes. To ensure an accurate comparison, 9 and 27 genes that were not assayed by the Affymetrix Mouse 1.0ST array were excluded from the GO and MP gene sets, respectively. Genes with relevant GO and MP annotations that were not included in the mDLCS because they failed the variance threshold were also removed from the comparison (83 and 496 genes, respectively). The final gene list for comparison to the mDLCS contained 117 lung development genes and 553 respiratory system phenotype genes. The mDLCS contained 87 of 117 “lung development” genes (74%) and 381 of 553 “respiratory system phenotype” genes (69%); 57 of these genes captured by the mDLCS were annotated with both “lung development” and “respiratory system phenotype.” The lists genes with GO and MP annotations used for this analysis are provided in Data S9.

The mDLCS contained 4,596 genes that are not currently annotated with “lung development” and 4,203 not annotated as “respiratory system phenotype.” Some of these are well-known development genes that have not been annotated specifically to lung development terms by GO curators. Others, however, represent novel lung development genes. For example, we discovered that four members of the dihydropyrimidinase-like 2 family (Dpysl2, Dpysl3, Dpysl4, Dpysl5) are expressed during embryonic (EMB) and alveolar (ALV1-4) stages, suggesting a role for these genes in pulmonary innervation during organogenesis and postnatal alveolarization. Although these genes have not previously been reported as lung development genes, the expression levels of the human homolog of Dpysl2 has been reported to be significantly upregulated in the tumors of small cell lung cancer patients (Taniwaki et al., 2006).

Strain-specific patterns of gene expression during lung development

As reported above, PC 4–10 lacked significant correlation among the strains and regression modeling revealed significant strain effects (Table 1). Of 20 calculated strain terms, 11 were significant (FDR < 0.1), including at least one term for each of PC 4–10. In contrast, only four PCs (5–6, 9–10) had a significant interaction (strain*stage) effect for one or more developmental stages; of 180 calculated interaction terms, only 7 were judged significant (FDR < 0.1). Thus, the dominant strain-specific expression patterns captured by PC 4–10 are stage-independent, with only smaller stage-specific variations observed.

Table 2 Summary of strain-dependent expression patterns for lung development transcriptional profiling in three inbred mouse strains.

Four classes of strain-dependent gene expression account for the strain effects observed on PC 4–10. The percentage of genes in each of the four classes is based on total number of genes with significant strain-differences by Tukey HSD (Q < 0.05). The patterns within classes reflect the expression levels of genes in the outlier strain relative to the indistinguishable strains. The percentages of genes for each pattern are based on the total of genes in the parent class.

Classes (4)	Patterns	
C3H≠AJ≈B6	C3H > AJ ≈ B6	C3H < AJ ≈ B6	
223 genes (28.6%)	89 genes (39.9%)	134 genes (60.1%)	
B6≠AJ≈C3H	B6 > C3H ≈ AJ	B6 < C3H ≈ AJ	
349 genes (44.7%)	142 genes (40.7%)	207 genes (59.3%)	
AJ≠B6≈C3H	AJ > B6 ≈ C3H	AJ < B6 ≈ C3H	
152 genes (19.5%)	85 genes (55.9%)	67 genes (44.1%)	
C3H≠AJ≠B6			
57 genes (7.3%)			
Notes.

AJ A/J

C3H C3H/HeJ

B6 C57BL/6J

PCs 4–10 were evaluated for patterns of significant strain effects, specifically how the patterns differentiate each strain from the others. Although nineteen possible patterns of strain-dependent expression are possible with three strains, the results from regression modeling demonstrated that the following four classes account for nearly all strain-dependent effects (Fig. 3):

• C3H different from indistinguishable pair AJ/B6 (PC4)

• B6 different from indistinguishable pair AJ/C3H (PC6, PC7)

• AJ different from indistinguishable pair B6/C3H (PC8)

• All strains different (PC5, PC9, PC10)

Regression modeling of gene expression as a function of strain, and subsequent Tukey HSD testing (Q < 0.05), quantified the genes that significantly contributes to each class (Table 2). Gene expression associated with lung development is most dissimilar between B6 and AJ or C3H mice; a 40/60 ratio of genes that are up versus down regulated in B6 relative to AJ/C3H was also observed. Differences between C3H vs AJ/B6 and AJ vs C3H/B6 exhibited 40/60 and 60/40 ratios of up versus down regulated genes, respectively. Differences between all strains were less common; only 57 strain-dependent genes (7.3%) were differentially expressed between all three strains. Strain-dependent gene expression was validated by qPCR (Fig. S11) in four target genes (Fggy, Saa3, Wnt11, Wif1). The qPCR results confirmed the trends observed in the array data despite the variation observed among biological replicates (Table S2). All strain effects for Saa3 and Wif1 were confirmed by qPCR; that is, Saa3 expression is greatest in C3H and indistinguishable between AJ and B6 whereas Wif1 expression is greatest in AJ and indistinguishable between B6 and C3H. The qPCR results for Fggy were consistent with the trend of AJ < B6 < C3H; however, the expression differences were not statistically significant between AJ and B6. Similarly Wnt11 expression was greatest in B6 but that B6 was not statistically significant different from the AJ strain. These results suggest that the general trends of strain-dependent expression captured by the arrays are robust but that expression of strain-differences for individual genes should be validated by qPCR or single cell sequencing prior to experimental follow up to investigate the biological significance of these differences.

C3H different from AJ and/or B6

Gene expression patterns for C3H were significantly different from AJ or B6 in nearly all strain-dependent components (PC 4–5, 8–10). Differences in the expression of genes associated with cell migration, chemotaxis, and immune system function contribute to this pattern. The induction of twelve genes (Amica1, Cd24a, Ccl3, Ccr3, Csf3r, Cxcl13, Cxcr2, Nckap1l, Ptafr, Retnlg, Saa3, Spp1) associated with immune system chemotaxis was observed in C3H (relative to AJ or B6) during late postnatal stages of alveolarization ALV3 and ALV4 (Fig. S7). Furthermore, 20 genes associated with chemotaxis (GO:0006935) follow a similar pattern distinguishing B6 from C3H. These differences in chemotactic signaling may be partly explained by strain-dependent differences in respiratory immune cell populations; specifically CD103+ dendritic cells, natural killer cells and/or TCR γδ+ T lymphocytes (Hackstein et al., 2012). Alternatively, increased expression of chemotactic factors during later stages of alveolarization and vascular remodeling may suggest an extended period of lung growth in C3H mice, which are known to have significantly larger lungs (by volume) than either B6 or AJ (Reinhard et al., 2002; Soutiere, Tankersley & Mitzner, 2004).

B6 different from AJ and/or C3H

Components distinguishing B6 from C3H and AJ (PC6 and PC7) have opposite strain effects yet highly similar temporal profiles (stage effects) suggesting they capture four sets of genes (one positive set and one negative set per PC) that are modulated in sync throughout lung development; two of these gene sets (PC6pos and PC7neg) are expressed higher in B6 whereas the other two (PC6neg and PC7pos) are expressed higher in AJ and C3H (Fig. 3). Characteristic genes contributing to the B6high signal (PC6pos and PC7neg) were enriched for cellular component ECM, and biological processes related to branching morphogenesis and neurogenesis. Characteristic genes contributing to the B6low signal (PC6neg and PC7pos) were enriched for biological processes lung alveolus development, respiratory tube development, lung cell differentiation, and neurogenesis. Regression modeling of genes involved in neurogenesis revealed 58 significant genes that were differentially expressed between B6 and C3H or AJ; eight of these genes (Fig. S8) also had significant stage*strain effects differentiating expression in B6 from C3H or AJ during the embryonic (EMB) stage of development (Isl1, Foxp1, Nefl, Nefm, Kif5c, Epha4, Sema3d, Nr2f1). These results suggest that genes involved in branching morphogenesis and ECM function of the developing lungs are expressed at higher levels in B6 mice than C3H or AJ mice. Conversely, genes involved in alveolar development and cellular differentiation are expressed at lower levels in n the developing lungs of B6 mice compared to C3H or AJ mice.

AJ different from B6 and C3H

Gene expression patterns distinguishing AJ from B6 or C3H were detected on PC8. Genes contributing to this pattern (Fig. S9) were associated with a broad range of biological functions including metabolism and cellular growth (Ggh, Pnpo, Igfbp3, Itln1, Tbcb), response to hypoxia (Chchd2), and the heat-shock degradation pathway (Dnajc10). Several of these genes also play peripheral roles in neuronal growth and/or physiology, suggesting that AJ mice may exhibit differences in pulmonary innervation or function compared to C3H or B6 mice.

Because the three mouse strains included in this study are known to exhibit differences in susceptibility to induced lung fibrosis (B6 > AJ > C3H) we investigated the strain-dependent expression levels of genes in the pro-fibrotic Wnt signaling pathway (Chilosi et al., 2003; Douglas et al., 2006; Konigshoff et al., 2008). A key regulator of epithelial-mesenchymal crosstalk (Wnt2b) exhibits increased expression in B6 mice relative to AJ or C3H throughout most of lung development. Further investigation of Wnt family genes identified increased postnatal expression of Wnt5a, Wnt10b, and Wnt11 in B6 mice relative to AJ/C3H as well as decreased expression of Wnt-inhibitory factor 1 (Wif1) in B6 compared to AJ/C3H (Fig. S10A ). Regression analysis confirmed significant strain-differences (P < .0001) in the expression of each of these Wnt-related genes as well frizzled receptors 3, 4, and 6 (Wnt2b, Wnt10b, Wnt11, Wif1, Fzd3, Fzd4, Fzd6). The expression profiles of 12 other genes associated with pulmonary fibrosis (MP:0006050) were not correlated between strains (Fig. S10B) as determined by regression analysis. Our analysis identified several additional genes with both strain-specific expression patterns and reported roles in pulmonary fibrosis or immune function, including Cysltr2 (B6 > AJ > C3H) (Beller et al., 2004), Asah1/Asah2 (B6 > AJ = C3H) (Dhami, He & Schuchman, 2010), Hck (B6 > C3H > AJ) (Ernst et al., 2002), Gas5 (B6 > AJ = C3H) (Song et al., 2014), and Cpa3 and Mcpt4 (C3H = AJ > B6) (Paun & Haston, 2012). These results suggest a potential role for Wnt signaling in the strain-dependent modulation of vascularization during alveolar development, as well as shed light on putative signaling pathways and immune modulators involved in conferring strain-dependent resistance susceptibility to pulmonary fibrosis.

Comparative analysis of the mouse and human embryonic developing lung characteristic subtranscriptomes

To investigate the genomic elements of lung development that are conserved between human and mouse we performed PCA on a reduced set of murine samples (E12.5–E16.5) that corresponded to the time points assayed in a previously published human Developing Lung Characteristic Subtranscriptome dataset containing 3,209 genes (Kho et al., 2010). We generated a prenatal mDLCS using the union of rank-ordered genes from PC1-3 that resulted in a set of 3,077 genes. Most of the genes (2,302) in the prenatal mDLCS overlapped the mDLCS that included all 26 developmental time points.

Homologous human genes could not be identified for 464 of the prenatal mDLCS genes and homologous mouse genes could not be identified for 201 of the hDLCS genes, leaving 2,634 genes in the prenatal mDLCS for comparison to the 2997 hDLCS gene set. The genes from the prenatal mDLCS genes for which no human homolog could be found included predicted genes, microRNAs, long non-coding RNAs (lncRNAs), and species-specific immune-related proteins (e.g., major histocompatibility complexes). There were 771 genes shared in common (RF = 2.3, P < 2.4 × 10−134) between the two datasets following the conversion of the mouse genes to human homologs; 2,226 genes were unique to the hDLCS and 1,861 genes were unique to the prenatal mDLCS (Fig. 5).

Figure 5 Comparison of mouse and human embryonic Developing Lung Characteristic Subtranscriptome (DLCS) gene sets.

Schematic of workflow for comparing embryonic DLCS gene sets from mouse and human. Mouse genes were converted to human homolog gene symbols to facilitate comparison. Results of annotation enrichment analysis for the unique and overlapping genes are listed. Complete results available in Data S9.

There are two primary reasons for the lack of overlap of specific genes between the mouse and human subtranscriptomes. First is the difference in the genes represented on the two gene expression Affymetrix array platforms (Mouse 1.0ST and the Human 133 Plus 2.0). Mouse homologs for 68 genes included in the hDLCS were not assayed by the Mouse 1.0ST array. Second, mouse homologs for 1,216 of the genes in the hDLCS lacked variance in expression during embryonic development and were removed at the variance-filtering step prior to PCA. These genes therefore, were not included in the prenatal mouse DLCS.

Although the overlap of individual genes between the hDLCS and mDLCS was small, the biological processes and pathways represented by the DLCS gene lists were similar. Both DLCS gene sets were enriched in genes involved in broad developmental processes associated with lung development including cell cycle, lung-specific metabolism, signal transduction, and a wide range of immune functions (Table 3).

Table 3 Summary of Reactome pathway enrichment results for the 771 genes represented in both mouse and human embryonic developing lung characteristic subtranscriptomes.

Entities refer to proteins, molecules, sequences, and other physical complexes associated with a given pathway in Reactome database; entities found are those associated with input gene sets and total entities refers to all entities in a given pathway. FDR represents multiple testing corrected p-values for enrichment.

	Entities found	Entities total	Percent total	FDR	
Cell cycle	92	608	15.1%	8.97 × 10−14	
Cell cycle checkpoints	22	169	13.0%	1.80 × 10−2	
ATR response to replication stress	8	39	20.5%	4.77 × 10−2	
Mitotic G1-G1/S phases	21	139	15.1%	4.46 × 10−3	
Pre-replicative complex activation	8	33	24.2%	2.76 × 10−2	
Mitotic G2-G2/M phases	23	141	16.3%	7.50 × 10−4	
Polo-like kinase-mediated events	9	23	39.1%	6.70 × 10−4	
S phase	16	140	11.4%	9.08 × 10−2	
M phase	51	316	16.1%	2.84 × 10−8	
Resolution chromatid cohesion	27	128	21.1%	1.83 × 10−6	
Metabolism	135	3,163	4.3%	1.00 × 101	
Erythrocyte gas exchange	4	16	25.0%	1.40 × 10−1	
Surfactant metabolism	7	52	13.5%	1.96 × 10−1	
Immune system	132	1,841	7.2%	6.69 × 10−2	
Adaptive immune system	75	1,002	7.5%	1.18 × 10−1	
TCR signaling	25	145	17.2%	1.63 × 10−4	
MHC-II antigen presentation	30	141	21.3%	3.20 × 10−7	
Costimulation by CD28 family	24	96	25.0%	5.42 × 10−7	
PD-1 signaling	17	45	37.8%	3.20 × 10−7	
Cytokine signaling in immunity	68	678	10.0%	5.66 × 10−4	
Interferon signaling	35	289	12.1%	2.43 × 10−3	
Interferon gamma signaling	21	138	15.2%	3.77 × 10−2	
Antiviral mechanism IFN-stimulated	13	83	15.7%	3.60 × 10−2	
Interleukin signaling	31	358	8.7%	1.56 × 10−1	
DAP12 signaling	33	364	9.1%	9.02 × 10−2	
Developmental biology	86	904	9.5%	8.68 × 10−6	
Axon guidance	66	568	11.6%	8.68 × 10−6	
L1CAM interactions	16	124	12.9%	5.66 × 10−2	
NCAM signaling (neurite out-growth)	29	298	9.7%	6.69 × 10−2	
Semaphorin interactions	11	73	15.1%	6.69 × 10−2	
Signaling by robo receptor	8	38	21.1%	4.44 × 10−2	
Signal transduction	211	2,710	7.79%	3.42 × 10−1	
PIP3 Activates AKT sgnaling	15	121	12.4%	6.69 × 10−2	
IRS-mediated sgnaling	28	306	9.2%	1.21 × 10−1	
SCF-KIT sgnaling	34	339	10.0%	3.94 × 10−2	
EGFR signaling	38	376	10.1%	2.76 × 10−2	
FGFR signaling	33	356	9.3%	7.44 × 10−2	
PDGF signaling	38	392	9.7%	3.88 × 10−2	
ERBB2 signaling	33	360	9.2%	8.04 × 10−2	
ERBB4 signaling	34	345	9.9%	4.77 × 10−2	
Rho GT pase signaling	64	418	15.3%	1.52 × 10−9	
NGF signaling	43	478	9.0%	6.29 × 10−2	
Wnt signaling	35	326	10.7%	1.74 × 10−2	
IGF1R signaling	29	311	9.3%	9.40 × 10−2	

The 2,226 genes unique to the hDLCS set were significantly enriched for cell cycle and DNA repair processes, which may reflect differences in tissue quality, harvesting, and/or processing between studies. The 1,861 genes represented only in the mDLCS were enriched for high-level biological processes (cell cycle, DNA replication) as well as ECM organization, specifically non-integrin membrane-ECM interactions (LAMA3/4, LAMB1, COL4A1/4A2, COL2A1), ECM proteoglycans, degradation of the ECM, and integrin cell surface interactions. The set of genes associated with enrichment of ECM organization includes alpha/beta integrins (ITGA6/8, ITGB6), laminins (LAMA3/4, LAMB1), collagens (COL1A2, COL2A1, COL4A1/2, COL6A1/2/3, COL14A1, COL26A1), adhesion molecules (ICAM1/2, JAM2/3, VCAM1), cathepsins (CTSB, CTSS, CTSV), and metallopeptidases (ADAMTS5/9, MMP14/19). These results suggest that underlying differences in ECM remodeling and/or composition may exist between the mouse and human prenatal developing lung microenvironment.

The previous human lung transcriptome study by Kho et al. (2010) revealed evidence of a novel pseudoglandular substage between the 13th–17th weeks of human lung development (corresponding to E15.5 in mouse). Our mouse embryonic transcriptome data did not recapitulate this finding. Although the plots of PC sample scores for PC1 and PC2 in the B6 mouse strain show some similarity to the plots for the human data, strain-dependent variance of the PCA sample scores and the variance in gene expression for the B6 E15.5 time points complicate the comparisons. Only three of the more than 50 genes reported to be differentially expressed between the early and late pseudoglandular (PSG) time points in humans are differentially expressed between E14.5 and E15.5 in mouse: Sulf1, Muc1, and Sftpc. The lack of concordance between the mouse and human data is likely related to the sampling of lung development time points in mouse. The PSG sub stage defined in humans lasts approximately five weeks whereas the comparable developmental time in mice occurs over several hours. To definitively address the presence of a novel pseudoglandular stage in mouse would require sampling lung development at a much finer temporal scale (hourly) than was performed for the current study (one sample per day). An alternative explanation to the lack of a PSG substage in our murine dataset may lie in the fact that only male mice were used in this study; recent work in humans has revealed gender-based differences in transcriptional modulation surrounding this substage (Kho et al., 2015).

Discussion and Conclusion

In this report we present the results of the most comprehensive characterization of gene expression during normal murine lung development to date. The dataset consists of gene expression measured at 26 time points from E9.5-P56 in three common inbred strains of mice. Using a combination of Principal Component and least squares regression analysis we identified strain independent and strain dependent patterns of genome wide transcription during pre- and post-natal development. These analyses provide significant impact for both basic and translational research into mammalian lung development through the generation of a high-resolution molecular framework of murine lung development, comparative genomic analyses of human and mouse lung development, and the identification of putative pathways associated with respiratory pathology.

Principal components 1–3 suggest that lung development utilizes superimposed periodic patterns of transcriptional control

The plots of PC sample scores for the first three principal components across the sampled developmental time points revealed distinct periodic patterns of gene expression during lung development (Fig. 2) similar to those reported recently for developmental gene expression in the nematode, C. elegans (Hendriks et al., 2014). The temporal expression pattern for PC1 divides the developmental timeline in two segments: (1) embryonic, pseudoglandular, and canalicular stages versus (2) saccular, alveolar, and mature stages. The global trend is partially reversed in the second and third alveolar stages. The singular transition point from embryonic to post-natal pattern that occurs on PC1 between the canalicular and saccular stages supports the previous assertion that the saccular stage of lung development is a critical period of preparation for the switch to breathing air in mice (Kho et al., 2009). PC2 captures a developmental pattern of gene expression that has two major changes, occurring between EMB and PSG stages and then reversing again between ALV3 and ALV4 stages. As previously observed in Kho et al. (2009), PC2 effectively distinguishes stages temporally nearer to birth (PSG, CAN, SAC, ALV1, ALV2, and ALV3) from those more distant (EMB, ALV4, and MAT). Finally, PC3 captures a pattern with four distinct phases, with transitions found between PSG and CAN, SAC and ALV1, and ALV4 and MAT stages of development. In effect, the PC3 expression profile from early embryo to birth is repeated from birth to maturity. Somewhat strikingly, where PC2’s expression profile is inverted in mature lungs compared to late embryological development, PC3’s profile matches that of late development. The regulatory control and consequences of these periodic patterns of increasing frequency are topics for future inquiry.

Alveolarization in mouse has four distinct transcriptional stages suggesting waves of vascularization and innervation

A regression analysis of the principal components identified nine distinct stages of lung development represented by the gene expression data. Five of these stages were consistent with previous definitions of lung development (embryonic, pseudoglandular, canalicular, saccular, and maturation following alveolarization). In contrast to previous studies that proposed alveolar development as an extension of the saccular stage (Bonner, Lemon & You, 2003) or as two stages (Mund, Stampanoni & Schittny, 2008; Schittny, Mund & Stampanoni, 2008) our data suggest subdivision of postnatal alveolarization into four distinct stages (Fig. 4).

Differential gene expression analysis between successive stages of development revealed two distinct waves of angiogenesis during alveolarization (Fig. 6). Multiple angiogenic regulators, including Vegfa, Adora2b, Adrb2, Lgals3, Tgfb1 and Ang, were significantly induced in the first alveolar stage (ALV1), relative to the saccular stage (SAC), and again in the fourth alveolar stage (ALV4) relative to the third (ALV3). Further support of the two-phase pattern of angiogenesis is found in the stage-specific (ALV2/ALV3) induction of genes associated with negative regulation of angiogenesis (Thbs2, Agt) and/or vascular stabilization (Angpt1, Angpt2, Serpine1, Igf1). Seven of these vascular stabilization genes were also expressed in the mature lung (MAT) suggesting that there are periods of angiogenic “rest” during postnatal maturation and adult homeostasis (Fig. S3). A similar dual-phase pattern of induction (ALV1 and ALV4-MAT) was detected among genes known to be involved in alveolarization and neurogenesis (Fig. S4 and S5), suggesting extensive genomic coordination between vascular, neurogenic, and alveolar processes in the postnatal lung development.

Figure 6 Post-natal expression patterns of genes associated with alveolarization and angiogenesis.

Schematic of anti-correlated patterns of gene expression levels for genes associated with alveolarization and angiogenesis (solid line) versus negative regulation of angiogenesis (dotted line). The patterns illustrate functional genomics basis of the four stages of alveolar development proposed in this study.

PCs 1–3 define a “Developing Lung Characteristic Subtranscriptome” in mouse

We combined the genes from the principal components that had no significant strain dependent effects into a murine Developing Lung Characteristic Subtranscriptome (mDLCS). The mDLCS represents the core set of genes expressed in specific temporal patterns during lung development in the mouse, independent of strain-specific genetic effects. When we compared our mDLCS to lists of genes known previously to be involved in murine lung development we determined that the majority of these genes were represented in the mDLCS. There were two primary factors that excluded previous lung development genes from our mDLCS. First, some genes did not pass the threshold we used to exclude genes from the PCA/regression analysis that defined the mDLCS. For example, 7/20 GO-annotated genes and 67/115 MP-annotated genes that were not included in the mDLCS were near the threshold criterion and would have been included with only minor changes in the threshold requirements. Second, some genes were not included because they displayed strain specific expression patterns and therefore were captured in PC4–10 instead of PC1-3.

Strain specific gene expression during lung development

The inclusion of multiple inbred strains of mice in our study gave us the opportunity to identify strain-dependent patterns of gene expression. Regression modeling revealed the majority of these patterns represent global differences between strains that are independent of developmental stage. The four primary patterns of strain-dependent expression that we uncovered were largely characterized by expression differences in genes associated with ECM composition, lung development, neurogenesis, immune system chemotaxis and function, and profibrotic pathway signaling. Differences in leukocytic composition between strains may explain the strain-dependent differential expression of chemotactic factors; however further investigation is necessary to shed light on putative strain-dependent differences in pulmonary ECM composition, alveolar versus bronchiolar development, and pulmonary innervation.

The expression profiles of several immune-related or profibrotic genes, including those that influence inflammatory response, ECM remodeling through MMPs, and epithelial-mesenchymal crosstalk, were among the genes showing strain differences and may help explain the known differences in susceptibility for asthma and pulmonary fibrosis between B6 (susceptible) and AJ (resistant) or C3H (very resistant). Further investigation of the profibrotic Wnt signaling pathway also identified gene expression differences between B6, AJ, and C3H mice that may explain phenotypic differences in fibrotic susceptibility between strains. We observed increased expression of mast-cell secretory factors (Cpa3, Mcpt4) in AJ and C3H mice compared to B6 immediately after birth, which may reflect differences in pulmonary inflammatory response between strains and is consistent with experimental data showing elevated mast cell numbers and exacerbated alveolitis in AJ and C3H relative to B6 following radiation (Haston, 2012).

Comparison of mouse and human embryonic lung development

Our comparison of a murine embryonic DLCS to the previously published human embryonic DLCS revealed only moderate overlap at the gene level. Among the genes we identified in common between the mDLCS and hDLCS were key cell differentiation factors (AGER, ABCA3, GPRC5A, EGFL6), primary surfactant-/mucin-associated proteins (ADGRF5, CTSH, MUC1, SRFP1/2, SFTPA1/2, SFTPB, SFTPC), or major structural components, including adhesive junction-associated proteins (CDH1, CADM1, MATR3, MLLT4, SCAPER) and tight junction-associated claudins (CLDN5, CLDN18) (Table S2). The lack of substantial overlap of the DLCS gene lists at the level of individual genes may be due, in part, to the differences in the array platforms used in the two studies. Mouse lung expression was assayed using Affymetrix 1.0 ST gene arrays where probesets are largely targeted to protein coding regions. In sharp contrast, the human lung expression dataset used U133 Plus 2.0 arrays, which has probesets targeted primarily to 3′untranslated regions (UTRs). Prior studies have frequently identified developmental and cell-specific transcript isoform variation specifically in the 3′-UTR. The two array designs will have different sensitivity to such variations, potentially resulting in very different measurements, even with common transcript expression profiles, and subsequently changing the identification of genes with enough variation for inclusion into downstream analysis.

In contrast to the modest overlap of individual genes, the enriched annotations associated with the genes in the mDLCS and hDLCS revealed many shared functional properties. Both subtranscriptomes were enriched in genes involved in axon guidance, specifically through L1CAM/NCAM interactions and signaling through semaphorins/Robo, suggesting pulmonary neurogenesis events during prenatal development of the lung are highly conserved between mouse and human. Several signaling transduction pathways were also enriched among conserved elements of the mDLCS and hDLCS, including signaling through insulin receptors, protein kinases, receptor tyrosine kinases, Wnt, and Rho GTPases. Developmentally associated growth factor mediated signaling was also conserved including epidermal growth factor, fibroblast growth factor receptor, platelet-derived growth factor, nerve growth factor, and stem cell factor.

Both hDLCS and prenatal mDLCS gene sets were also enriched in general immune system cytokine signaling, including interferon- and interleukin-signaling, as well as adaptive immune system, including TCR signaling, MHC-II antigen presentation, and PD-1 signaling. This was surprising since both subtranscriptomes are based entirely on pre-birth development time points and suggests an unexpected role for adaptive immune system elements in prenatal lung development. Notable differences in annotation enrichment results between the mouse and human subtranscriptomes were also detected. For example the mouse embryonic DLCS gene set was enriched for genes involved in ECM organization whereas the human lung DLCS (but not murine) showed enrichment of genes involved in protein biosynthesis and protein modification.

The data reported here expand the current knowledge of genome wide temporal changes in gene expression during the complex process of murine lung development, revealing novel stages of postnatal alveolar development. The data generated for this study complement existing transcriptomics data sets for murine lung development such as those that are available from the LungMAP resource (http://www.lungmap.net/). The normal mouse lung gene expression time course data available from LungMAP includes only two embryonic and two post-natal time points for a single strain (C57BL/6). Our data support a much finer-grained assessment of the temporal dynamics of gene expression across lung development and the ability to identify possible strain-specific differences in expression. In contrast to our study, the LungMAP resource includes some data for cell-specific expression (from sorted endothelial cell, myeloid cell, and type II pneumocytes) at different development time points. These cell-specific expression data can be integrated with our data to develop hypotheses about which cell types may be contributing to the tissue level gene expression patterns observed in our data. Although a full treatment of combining the data set reported here with LungMAP data is beyond the scope of this report, the expression data for the angiogenin (Ang) gene serves as an example of the complementary nature of these data resources. The general pattern of increased expression of Ang over development is observed in both data resources (see www.lungmap.net and lungdevelopment.jax.org). However our data reveal that the relative level of expression of Ang is much higher in C57BL/6J than in A/J or C3H/HeJ. This strain-specific difference is not possible to detect in the existing LungMAP data. The cell-specific expression data from LungMAP suggests that type II pneumocytes are contributing most to the increased expression of angiogenin between postnatal day 7 and day 28. This testable hypothesis would not be possible to formulate using our data set alone.

Strain-specific differences in gene expression patterns were identified that serve as the basis for future investigations into the genetic basis for well-known differences in adult lung physiology and respiratory disease susceptibility among inbred mouse strains. Comparison of developmental gene expression data between mouse and human highlighted conserved biological processes as well as organismal differences that may be important for using the mouse as a model for human respiratory biology and disease. The raw data files and processed data for this study can be accessed at NCBI’s Gene Expression Omnibus (GEO) data archive (GSE74243). Expression patterns across stages (and time points) of lung development for all genes measured by the Mouse Affymetrix Gene 1.0ST array can be queried, visualized, and downloaded at http://lungdevelopment.jax.org/.

Supplemental Information

Data S1 Genotyping protocol for determining the sex of mouse embryos

Click here for additional data file.

Data S2 Summarized gene expression values for lung development in three inbred mouse strains

The mouse lung development gene expression data were filtered to remove genes that had low levels of temporal expression variation.

Click here for additional data file.

Data S3 Results from principal component analysis of lung development gene expression: gene loading values

Click here for additional data file.

Data S4 Results from principal component analysis of lung development gene expression: principal component sample scores

Click here for additional data file.

Data S5 Results of differential gene expression analysis across stages of lung development using SAM

Click here for additional data file.

Data S6 Results of annotation enrichment analysis for the mouse Developing Lung Characteristic Subtranscriptome (mDLCS)

Click here for additional data file.

Data S7 Explanation of annotation term enrichment analysis

Click here for additional data file.

Data S8 Genes identified in previous transcriptional profiling studies of mouse lung development

The genes included in these lists were compared to the genes included in the mouse Developing Lung Characteristic Subtranscriptome (mDLCS) defined in this study.

Click here for additional data file.

Data S9 Mouse genes annotated with ontology terms for respiratory phenotype and lung development

The genes included in these lists were compared to the genes included in the mouse Developing Lung Characteristic Subtranscriptome (mDLCS) defined in this study.

Click here for additional data file.

Figure S1 Relationships between gene loading values and PCA score

(A) Plots of gene loading values for PC 1–10. (B) Plot of PCA scores for PC1 across lung development time points sampled in this study highlighting a dramatic shift in gene loading values just before birth. (C) Mean expression levels of two representative genes with inverse loading values from PC1. Black bars, expression of Nasp (positive loading value); light grey bars, expression of Gpr116 (negative loading value). Error bars reflect one standard error from the mean. Complete PCA results of gene loading values and PCA scores found in Data S3 and S4, respectively.

Click here for additional data file.

Figure S2 Principal component analysis reveals strain-dependent transcriptional patterns throughout lung development Plots of PCA scores by strain and stage for principal components showing strain-dependent correlations

Click here for additional data file.

Figure S3 Genes associated with pulmonary vascularization or angiogenic rest are differentially expressed during postnatal lung development Plot of Z-scores (y-axis) for gene expression levels of selected genes associated with pulmonary vascularization across stages of lung development (x-axis)

Each error bar is constructed using 1 standard error from the mean. The patterns of expression illustrate the concept of periods of “angiogenic rest” during alveolarization. Potent angiogenic factors (Vegfa, Tgfb1, Ang) and known regulators of pulmonary vascularization (Adora2b, Adrb2, Lgals3) have elevated expression levels at alveolar stages (ALV1 and/or ALV4). Genes associated with the negative regulation of angiogenesis (Thbs2, Agt) and vascular stabilization/maturation factors (Angpt1, Angpt2, Serpine1, Igf1) show an inverse relationship of expression levels.

Click here for additional data file.

Figure S4 Genes associated with alveolarization are differentially expressed across multiple postnatal alveolar stages

Plot of Z-scores (y-axis) for gene expression levels of selected genes associated with lung alveolus development. Each error bar is constructed using 1 standard error from the mean. Transcription factors (Hopx, Nkx2-1, Errfi1), growth factors (Fgfr2, Vegfa) and genes involved in pulmonary surfactant production (Sftpd, Abca3, Gpr116, Napsa) are differentially expressed between ALV1-ALV2 and/or ALV3-ALV4.

Click here for additional data file.

Figure S5 Genes associated with axon guidance are differentially expressed across multiple stages of postnatal alveolarization

Plot of Z-scores (y-axis) for gene expression levels of selected genes associated with axon guidance and neurogenesis. Each error bar is constructed using 1 standard error from the mean.

Click here for additional data file.

Figure S6 Expression profiles of known and novel cell differentiation markers throughout lung development

Heatmap showing the relative expression of 35 genes previously associated with cellular differentiation in the lung. Each expression profile is relative to the average expression of that gene across all time points. Solid blue indicates 1.5-fold decrease relative to average; yellow indicates 1.5-fold increase. Gene loading values (PC1-3) are shown to right of heatmap; dark red shading indicates that a gene is within the top 5% of contributors to that respective PC; light red or dark red squares indicate genes that were captured by the mDLCS.

Click here for additional data file.

Figure S7 Genes associated with chemotaxis are differentially expressed between strains during postnatal alveolarization

Plot of Z-scores (y-axis) for gene expression levels of genes associated with chemotaxis. Each error bar is constructed using 1 standard error from the mean. (A) Genes associated with chemotaxis with higher expression levels in AJ or C3H relative to B6. (B) Genes associated with immune-related chemotaxis with higher expression levels in C3H relative to AJ or B6. Alveolar stages (ALV1-4) highlighted in red.

Click here for additional data file.

Figure S8 Genes associated with pulmonary innervation are differentially expressed in B6 relative to AJ or C3H during the EMB stage of lung development Plot of Z-scores (y-axis) for gene expression levels of genes associated with pulmonary innervation expressed lower in B6 relative to AJ or C3H

Each error bar is constructed using 1 standard error from the mean. Embryonic stage (EMB) highlighted in orange.

Click here for additional data file.

Figure S9 Genes that are differentially expressed in AJ relative to C3H or B6

Plot of Z-scores (y-axis) for gene expression levels of genes differentially expressed in AJ relative to C3H or B6. Each error bar is constructed using 1 standard error from the mean.

Click here for additional data file.

Figure S10 Expression profiles of genes associated with Wnt signaling and fibrosis across lung development with comparisons between strains Summary of strain effects and significant stage by strain effects for genes involved in Wnt signaling (A) and fibrosis (B)

Light red highlighting indicates genes that were included in the mDLCS; dark red highlighting indicates genes in the top 5% of contributors to a given PC. Heatmaps contrast expression levels in A/J (left) or C3H/HeJ (right) with C57BL/6J. Solid blue indicates a 1-fold increase in expression between strains; solid yellow indicates a 1-fold decrease in expression between strains.

Click here for additional data file.

Figure S11 qPCR validation of strain-specific differences in gene expression during lung development

ΔCt = cycle threshold, normalized to the geometric mean of three control genes (Actb, Rpl10, Rpl13a). Variation between biological replicates (detected by microarray) did not significantly impact trends of strain-variation when quantified by qPCR. Each error bar constructed using one standard error from the mean. Significant differences detected by Tukey multiple comparisons ANOVA between strains. ∗P < 0.05, ∗∗P < 0.01, ∗∗∗∗P < 0.0001.

Click here for additional data file.

Table S1 Postnatally induced toll-like receptor, lymphocyte antigen, chemokine receptor/ligand and interleukin genes with high loading values on PC1 The shift in gene expression patterns from prenatal development (EMB-CAN) to postnatal development (SAC-MAT) captured by PC1 (see Fig. S1) is characterized by expression changes in genes associated with immune system function

PCA-derived loading values (for PC1-3) also shown for each of these genes.

Click here for additional data file.

Table S2 Differences in strain-dependent gene expression at postnatal day 13 of murine lung development

Mean differences (Mean diff.) in relative expression units (A.U.) between strains analyzed by Tukey multiple testing corrected ANOVA. Results predominately conform to the trends of strain-dependent expression detected by microarray (shown next to each gene name on left) as seen by comparing qPCR significance to regression significance.

Click here for additional data file.

The authors are indebted to Dr. Marge Strobel and Ms. Rita Thibodeaux for their guidance and assistance with establishing the mouse colonies used for this study. Ms. Karen Hammond and Ms. Elsie Cough served as the Animal Care Technicians for the study. The gene expression assays were performed Ms. Sonya Kamdar and Ms. Sandy Daigle of The Jackson Laboratory Genome Technologies Scientific Services core. Dr. Nazira Bektassova assisted with the use of the deSNPer software. Mr. Keith Sheppard (The Jackson Laboratory Computational Sciences core) developed the database interface underlying the public gene expression visualization website. Mr. Jesse Hammer (The Jackson Laboratory MultiMedia group) assisted with the preparation of figures for this manuscript. This project was completed by KJB in partial fulfillment of his graduate dissertation research in the Graduate School of Biomedical Sciences and Engineering (GSBSE) at the University of Maine, Orono.

Additional Information and Declarations

Competing Interests

Author Contributions

Animal Ethics

Microarray Data Deposition

Data Availability

The authors declare there are no competing interests.

Kyle J. Beauchemin conceived and designed the experiments, performed the experiments, analyzed the data, contributed reagents/materials/analysis tools, wrote the paper, prepared figures and/or tables, reviewed drafts of the paper.

Julie M. Wells conceived and designed the experiments, performed the experiments, contributed reagents/materials/analysis tools, wrote the paper, reviewed drafts of the paper.

Alvin T. Kho contributed reagents/materials/analysis tools, reviewed drafts of the paper, secured funding.

Vivek M. Philip contributed reagents/materials/analysis tools, reviewed drafts of the paper.

Daniela Kamir analyzed the data, reviewed drafts of the paper.

Isaac S. Kohane reviewed drafts of the paper, secured funding.

Joel H. Graber conceived and designed the experiments, analyzed the data, contributed reagents/materials/analysis tools, wrote the paper, reviewed drafts of the paper.

Carol J. Bult conceived and designed the experiments, performed the experiments, analyzed the data, wrote the paper, reviewed drafts of the paper, secured funding.

The following information was supplied relating to ethical approvals (i.e., approving body and any reference numbers):

All animal work was performed in accordance with Jackson Laboratory Animal Care and Use Committee (ACUC) protocol 101011.

The following information was supplied regarding the deposition of microarray data: GEO GSE74243.

The following information was supplied regarding data availability:

The raw data has been supplied as Supplemental files.

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
