# Peer review of "Temporal dynamics of the developing lung transcriptome in three common inbred strains of laboratory mice reveals multiple stages of postnatal alveolar development"

_PeerJ, doi:10.7717/peerj.2318_

## Round 0.1 · original submission · Minor Revisions

This manuscript has received enthusiastic comments from all three reviewers. The preparation of the figures and clarity of the text is exemplary. The main comment I would like the authors to address is from Reviewer 2, who has drawn attention to the degree of variation (biological noise) between animals within strains. I agree with this Reviewer that additional data or analyses that demonstrated that within-strain reproducibility is greater than between strain-differences would improve an already impressive paper.

·

Basic reporting

This a clear well written manuscript, with strong background references to the literature.

Experimental design

This work presents a comprehensive analysis of the gene expression profile of the mouse lung during embryonic development and after birth in 3 different strains. The authors use principal component analysis to identify genes differentially expressed at different stages and between strains. The experimental approach is appropriate and well described.

Validity of the findings

The data presented are robust and conclusions drowned from the results are appropriate.

Additional comments

This article by Beauchemin et al. presents a comprehensive analysis of the gene expression profile of mouse lung developmental stages in 3 different mouse strains. The authors use a microarray affymetrix chip to generate expression data.
The data are available to the public through http://lungdevelopment.jax.org/ and constitute a great resource for researchers in the lung biology field.

In the discussion, could the authors comment on the differences in the data generated by their approach and the data available on http://www.lungmap.net/ where RNAseq expression data of lung specific cellular population in late stages of development are available?

The most surprising finding is the limited overlap between the genes expressed in the mouse embryonic lung and the corresponding time points in the human lung. The authors propose possible explanations for this observation that are well discussed.

Reviewer 2 ·

Basic reporting

none

Experimental design

Study design is comprehensive and thorough and technically defensible (see Validity of Findings).

Validity of the findings

There is a considerable variation between animals within strains that should be considered. This is relevant to the PCA comparisons between strains, and between murine and human data. The 'noise' within strains can be great, and therefore the power to detect unique gene sets can be significantly impacted. This study used very small numbers of embryos (3-4) or mice (3/grp/time point) and many time points so false discovery is possible as well as false negatives. Consider a qPCR validation of limited gene sets that were uniquely expressed by Strain X at timepoint Y for validation.

Additional comments

This is an excellent paper that addresses two important questions about strain and species (murine vs. human) effects on gene expression. It focuses on genes encoding proteins, for which there is precedent although the resolution of these experiments exceeds all past studies. There are many important findings concerning the strain differences in immune/inflammatory/ECM related genes which will be important to explore. Differences between murine and human transcriptome is striking suggesting that the processes are out of synchrony or relying on epigenetic control mechanisms which differ substantially. Overall the validity could be strengthened by verifying that within-strain reproducibility is greater than between-strain differences.

·

Basic reporting

Some elements of the article may not be accessible to a non-specialist audience, e.g. PCA and regression analysis of the data. But having said that, I feel that given the complexity of the analysis, this is unavoidable and does not deter from the value of the work.

Experimental design

No comments.

Validity of the findings

No comments.

Additional comments

The authors have created an exhaustive dataset covering transcriptional profiles of developing mouse lung. Furthermore, they created them for 3 different strains of the laboratory mice. I feel that this dataset will be extremely useful to the field and will aid our understanding of the processes controlling the lung development.

---

## Round 0.2 · accepted · Accept

Thank you for your comprehensive response to the reviewers' comments. This has significantly enhanced the paper.